# Optimizing clinical dosing of combination broadly neutralizing antibodies for HIV prevention

Bryan T. Mayer[1], Allan C. deCamp[1], Yunda Huang[1,2,3], Joshua T. Schiffer[1,4,5], Raphael Gottardo[1], Peter B. Gilbert[1,6], Daniel B. Reeves[1] *

1 Vaccine and Infectious Diseases Division, Fred Hutchinson Cancer Research Center, Seattle, Washington, United States of America, 2 Public Health Sciences Division, Fred Hutchinson Cancer Research Center, Seattle, Washington, United States of America, 3 Department of Global Health, University of Washington, Seattle, Washington, United States of America, 4 Department of Medicine, University of Washington, Seattle, Washington, United States of America, 5 Clinical Research Division, Fred Hutchinson Cancer Research Center, Seattle, Washington, United States of America, 6 Department of Biostatistics, University of Washington, Seattle, Washington, United States of America

* dreeves@fredhutch.org

**Data Availability Statement:** All relevant data are either within the manuscript and its Supporting Information files or can be found along with all code at https://github.com/bryanmayer/pkpd-bnab-

## Abstract

Broadly neutralizing antibodies (bNAbs) are promising agents to prevent HIV infection and achieve HIV remission without antiretroviral therapy (ART). As with ART, bNAb combinations are likely needed to cover HIV's extensive diversity. Not all bNAbs are identical in terms of their breadth, potency, and *in vivo* longevity (half-life). Given these differences, it is important to optimally select the composition, or dose ratio, of combination bNAb therapies for future clinical studies. We developed a model that synthesizes 1) pharmacokinetics, 2) potency against a wide HIV diversity, 3) interaction models for how drugs work together, and 4) correlates that translate *in vitro* potency to clinical protection. We found optimization requires drug-specific balances between potency, longevity, and interaction type. As an example, tradeoffs between longevity and potency are shown by comparing a combination therapy to a bi-specific antibody (a single protein merging both bNAbs) that takes the better potency but the worse longevity of the two components. Then, we illustrate a realistic dose ratio optimization of a triple combination of VRC07, 3BNC117, and 10–1074 bNAbs. We apply protection estimates derived from both a non-human primate (NHP) challenge study meta-analysis and the human antibody mediated prevention (AMP) trials. In both cases, we find a 2:1:1 dose emphasizing VRC07 is nearly optimal. Our approach can be immediately applied to optimize the next generation of combination antibody prevention and cure studies.

## Author summary

Some people living with HIV generate antibodies that can neutralize an extremely wide variety of HIV variants. Using these "broadly neutralizing antibodies" as drugs is an exciting development for HIV prevention and therapy. They are safe and well-tolerated, are

project. Additionally, the tool https://bnabpkpd.
fredhutch.org is freely available for use.

**Funding:** DBR gratefully acknowledges the support
of the NIAID through a K25 AI155224. Other
funding was provided by the National Institute of
Allergy and Infectious Diseases UM1 AI068614
[LOC: HIV Vaccine Trials Network] and UM1
AI068635 [HVTN SDMC FHCRC] to PG as well as
OPP1151646 via the Bill & Melinda Gates
Foundation to RG. The funders had no role in study
design, data collection and analysis, decision to
publish, or preparation of the manuscript.

relatively long-lasting, and hold the promise of one day being vaccine-induced. As broad as they are, early studies have shown that multiple antibodies will need to be combined to be most effective. Combinations can be complicated because some antibodies neutralize some variants better than others, and some last longer than others. We investigated how to balance these advantages and how to choose the ratios of antibodies to make the best combination drug. Our approach can immediately be used to optimize the coming generations of trials in humans.

## Introduction

Broadly neutralizing antibodies (bNAbs) are powerful agents that may become crucial for next generation HIV prevention [1]. Their utility is strengthened by their generally long half-lives compared to small molecule drugs, as well as the eventual promise of inducing bNAb production by vaccination [2,3].

The recent antibody mediated prevention (AMP) studies directly tested the hypothesis that the bNAb VRC01 could prevent HIV acquisition [4,5]. Viruses acquired by placebo recipients were more sensitive to neutralization by VRC01 than viruses acquired by VRC01 recipients. The prevention efficacy against sensitive viruses, defined as an 80% inhibitory concentration (IC80) < 1 μg/mL, was estimated at 75.4% (95% confidence interval 45.5 to 88.9%). More-resistant variants similarly infected placebo and control recipients. This study implies global HIV diversity [6]) remains beyond the breadth of any single current bNAb. As with antiretroviral treatment (ART) and pre-exposure prophylaxis (PrEP), combinations of products are likely needed [7–9].

Optimal bNAb combinations to achieve potency and breadth has been modeled previously [10,11]. The best bNAb combination to suppress viremia was also explored using a detailed model of viral fitness costs and bNAb escape [12]. However, these previous works do not include pharmacokinetic models or project *in vivo* potency. We previously integrated pharmacokinetic (PK) and multi-strain pharmacodynamic (PD) models to determine longitudinally varying potency of VRC01 and simulate the AMP studies [13]. Here, we extend and expand our PKPD model [13] into a combination bNAb study framework (**Fig 1**). In a triple antibody combination case study, we then apply the latest clinical correlates from non-human primate challenge studies [14] and the AMP studies [5] to best predict clinical efficacy from *in vitro* neutralization.

Our framework is designed to answer a design consideration for future studies with combination bNAbs for HIV: what is the optimal ratio of multiple antibodies to deliver? We show the optimal ratio can depend on many inputs and assumptions—precluding a one-size-fits-all solution. Instead, we provide a framework and a publicly available tool to determine the best dose plan given the specific antibodies, existing information about their interaction *in vivo*, and the PKPD outcome marker of interest for a proposed study. As knowledge of these components gets refined, the model framework will become more predictive.

## Results

### Pharmacokinetics (PK) for bNAb levels

The first component of the PKPD framework is the PK, describing concentrations of each antibody $i$ over time, $t$: $C_i(\theta_i, t, d_i)$ where $\theta_i$ are the bNAb specific PK parameters and $d_i$ is the initial dose (PK model in **Fig 1**). Individual initial dosing for each bNAb is then constrained

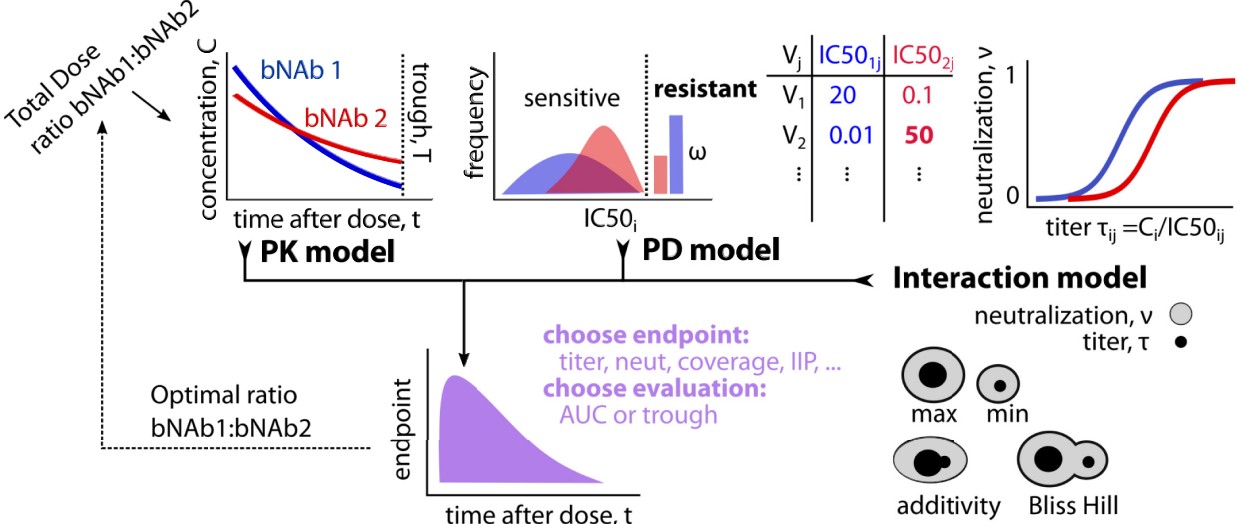

**Fig 1. PKPD model schematic for optimizing combination bNAb treatment against a genetically diverse pathogen like HIV.** The model incorporates: pharmacokinetics (PK), pharmacodynamics (PD), and interactions between broadly neutralizing antibodies (bNAbs). PK quantifies bNAb concentrations over time after administration. PD quantifies potencies at a given concentration for each antibody against many viral strains with sensitivity determined by $IC50_{ij}$, the level of the i-th drug needed to achieve 50% neutralization of the j-th viral strain–with some fraction ω of strains completely resistant. Titer, or the ratio of concentration to IC50 of each antibody against a certain strain, maps to neutralization proportion (0–1 scale) of viral infection events that are blocked. Interaction model includes taking either the worst (minimum) or the best (maximum) titer/neutralization between two products. Two more mechanistic interaction models combine titers (additivity) or neutralization (Bliss Hill), and generally mean combinations outperform the best single bNAb. Depending on the PKPD outcome measure of interest and when that measure is evaluated (throughout the study = AUC, at the low point = trough), we identify the optimal ratio of bNAbs.

by a total dose ($D = \sum_i d_i$). For simplicity, we assume a population-level fixed total dose and independent models of PK for multiple bNAbs (denoted $C_i(t)$ from here on). The model could be extended to implement individual-specific total dosing (e.g., bodyweight-adjusted) and joint, dependent models.

## Pharmacodynamics (PD) for bNAb potency

Two pharmacodynamic (PD) quantities are often used to discuss potencies given concentration: 50% inhibitory dose or dilution neutralization titer (ID50 Titer) and percent neutralization. Potency measures incorporate concentration and 50% inhibitory concentration (IC50) measurements across a panel of viruses (PD model in **Fig 1**).

Experimental neutralization titer (ID50), $\tau_{ij}(t)$, is a common measurement arising from titrated neutralization experiments. In practice, experimental ID50 represents a dilution factor applied to sera containing antibodies that reduces *in vitro* neutralization to 50%. Titer, and the similarly derived ID80, are important immunological endpoints that are proven correlates of protection [4,14]. Experimental titer can be theoretically predicted from the ratio of *i*-th drug concentration to *j*-th virus IC50 as

$$\tau_{ij}(t) = \frac{C_i(t)}{IC50_{ij}}$$ 

Eq 1

a relationship that has been empirically confirmed for single bNAbs [15,16]. As a potency measure, titer expresses the fold-relationship between concentration and viral IC50 as a measure of 'protection' against that virus.

Experimental *in vitro* neutralization for a single bNAb against a virus is also theoretically related to the titer (**Fig 1**). Percent neutralization (% neutralization) has the mechanistic

interpretation as the fraction of blocked cellular infection events by the *j*-th virus. Titer and % neutralization, *v*, can be related through the logistic Hill function (or median-effect equation) as follows

$$v_{ij}(t) = \left\{ 1 + \tau_{ij}(t)^{-h_{ij}} \right\}^{-1}. \qquad \text{Eq 2}$$

Neutralization requires an additional parameter, the 'Hill coefficient' $h_{ij}$, that describes the steepness of the neutralization curve. Through Eq 2, any generalized titer (e.g., ID80, ID99) can be predicted from the ID50 titer and a given Hill slope, where the Hill slope can be estimated from IC50 and IC80 measurements (see S1 Text). Using the CATNAP database [17] of IC50 and IC80 neutralization estimates for HIV virus/antibody combinations, we estimated the distribution of Hill slopes and generally found values near 1 (see Methods and S1 Fig). Henceforth in our analysis, and consistent with previous measurements [13], we set $h_{ij} = 1$ and it is dropped from equations. Under this assumption, the IC80 is theoretically predicted to be 4-fold higher than the IC50, and, subsequently, the ID80 is predicted to be 4-fold lower than the ID50 for single bNAb and virus combinations (see S1 Text for more details).

## bNAb interaction models

For bNAb combinations, we considered 4 PD interaction models. The first, Bliss-Hill independence (BH), is the best-case multiplicative interaction where bNAbs cover missing breadth of one another and co-neutralize strains, i.e., to establish infection virions must escape independent binding events from each antibody. BH is encouragingly observed from *in vitro* studies [10,18]. We also consider weaker cooperation with the additivity interaction model, where antibody effects are combined via mass action(10); i.e., the total titer is sum of individual titers. Finally, maximum and minimum models assume that the more or less potent antibody for each strain operates as a single product. The maximum interaction potentially represents a scenario where only the most potent bNAb neutralizes a given virus; however, outcome deviations between the maximum and the BH or additivity model also highlight where interactions improve neutralization due to combined coverage. On the other hand, the minimum model is mechanistically unrealistic but provides a boundary for the worst-case scenario where the combination regimen is only as strong as its weakest link, specifically penalizing poor combined coverage of viruses.

The interaction models are mathematically summarized in Table 1 and all derivations of combinations titers are included in the S1 Text. We extend interactions to include synergy in the bi-specific antibody case study, but do not consider antagonism among clinically viable bNAb combinations here.

Other options exist to quantify antibody potency, including instantaneous inhibitory potential (IIP [19]), the log-fold reduction in virus infectivity at a given concentration, which linearizes high neutralization on the log-scale (e.g., 99% neutralization -> IIP of 2, 99.9% -> 3)

**Table 1. Summary of equations for PD interaction models relating bNAb (*i*) to virus (*j*).** Formula for Bliss-Hill ID50 illustrated for 2-bNAb combinations only.

| PD Outcome | Bliss Hill (BH) | Additivity | Maximum | Minimum |
|---|---|---|---|---|
| Titer (ID50) $\tau_j(t)$, Eq 1 | $\dfrac{2\tau_{1j}\tau_{2j}}{-(\tau_{1j}+\tau_{2j})+\sqrt{(\tau_{1j}+\tau_{2j})^2+4\tau_{1j}\tau_{2j}}}$ | $\sum_i \tau_{ij}(t)$ | $\max_i[\tau_{ij}(t)]$ | $\min_i[\tau_{ij}(t)]$ |
| % Neutralization $v_j(t)$, Eq 2 | $1 - \prod_i[1 - v_{ij}(t)]$ | $1 - [1 + \sum_i \tau_{ij}(t)]^{-1}$ | $\max_i[v_{ij}(t)]$ | $\min_i[v_{ij}(t)]$ |

in the important range for ART efficacy [19].

$$\text{IIP}_{ij}(t) = -\log_{10}[1 - v_{ij}(t)] = \log_{10}[1 + \tau_{ij}(t)].$$  Eq 3

A generalized version of IIP when $h_{ij} \neq 1$ is described in the **S1 Text**.

Alternatively, the potency of an antibody combination can be quantified by its "viral coverage". Here, potency is dichotomized: a given virus is defined as "covered" if the continuous potency measure (% neutralization, IIP) is above a specified threshold value. Viral coverage is then the fraction of viral strains above the threshold. For example using % neutralization, for $n$ strains and a neutralization threshold $v^*$, we define the neutralization coverage fraction $f(t, v^*) = \frac{1}{n}\sum_{j=1}^{n} \mathcal{I}(v_{ij}(t) > v^*)$ where $\mathcal{I}$ is the indicator function equal to 1 if the inequality holds and 0 otherwise.

## Mathematical model for optimizing antibody combination doses

Finally, we summarize these measurements of potency over time, which we collectively term PKPD outcomes. We consider PKPD outcomes at trough (pre-specified final time) or throughout time (area under the curve, AUC) (**Fig 1**).

In practice, for a specified antibody combination, we obtain their PK parameters and the best estimate of their distribution of IC50s to a relevant panel of circulating viruses. We can then choose an interaction model and specify an outcome that we want to optimize. From this we uniquely determine the optimal ratio of the antibodies. Potential combinations of bNAbs—varying by their input PK and PD profiles—can then also be evaluated and compared via mathematical PKPD simulations at the optimal dosing ratios, which may be combination-dependent, as illustrated in the *in silico* studies below.

## Global sensitivity analysis

We performed a global sensitivity analysis varying all input PKPD model parameters (142,560 total combinations) to assess correlation between all PKPD outcomes and optimized dosing ratios (see **Methods**). Briefly, we varied one-compartment exponential PK models for each antibody summarized by their half-life $hl_i$. One bNAb was simulated to always have equivalent or better half-life than the other to avoid redundancy. We chose a log-normal distribution for IC50s for each bNAb parameterized by its mean $\mu_i$ and standard deviation $\sigma_i$ on the log10 scale, also allowing for a fraction $\omega_i$ that are completely resistant (infinite IC50). We also varied the ratio of doses $r$ and the total dose $D$. The ranges explored for each sensitivity analysis parameter are collected in **Table 2**.

The maximum and additive interaction were highly correlated (Spearman mean $\rho = 0.85$, range: 0.55–1). Henceforth the maximum is dropped and the additive model, which is more biologically established, is presented as a surrogate for both.

**Table 2. Parameter settings for global sensitivity analysis combining two bNAbs.**

| Parameter | Sensitivity analysis values |
| --- | --- |
| Initial dose (mg) | {150, 300, 600, 1200} |
| Half-life (days) | {7, 28, 42, 84} |
| Total simulated viruses | 500 |
| % viral resistance | {67, 33, 0} |
| Mean log10 IC50 (μg/mL) | {-3, -2, -1} |
| SD log10 IC50 (μg/mL) | {0.25, 0.5, 1} |

## Publicly available tool for ratio optimization

Any individual simulation from the results can be generated using the following R shiny app: https://bnabpkpd.fredhutch.org.

## PKPD outcomes cluster into categories

Using global sensitivity analysis output, we calculated Spearman correlations among all endpoints at trough (**Fig 2A**). By hierarchical clustering, we determined six main categories of outcomes (**Fig 2A**): All models with the minimum interaction (i.e., worst-case bNAb penalizing lack of combination viral coverage) and raw titer (ID50) endpoints for the non-minimum

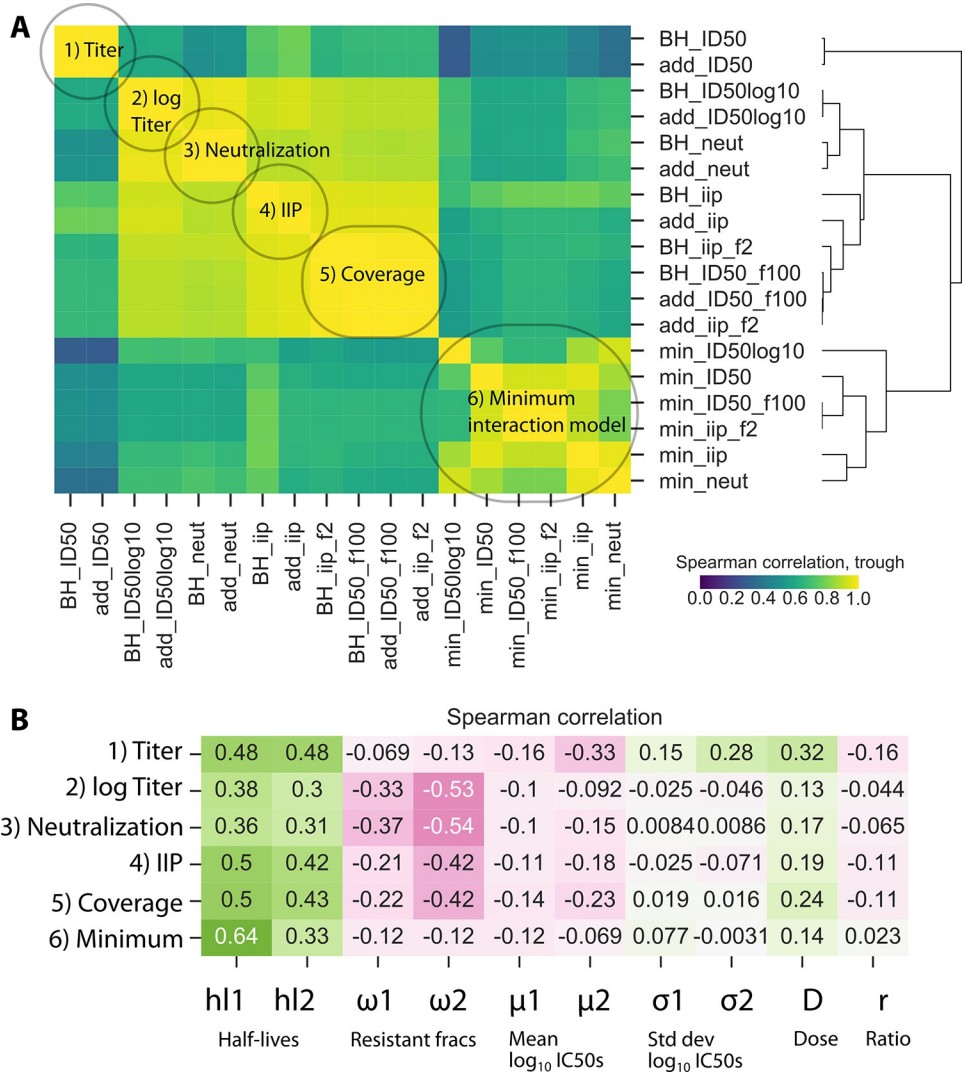

**Fig 2. Correlations among PKPD outcomes and between model parameters and outcomes.** A) Many metrics for PKPD outcomes are highly correlated (yellow in heatmap) and cluster into approximately 6 distinct categories: see labels. The minimum interaction was distinct across all outcomes. B) Of the 10 varied model parameters (**Table 2**), half-lives and resistant fractions had the largest impact on representative members of each of 6 categories from **A**. All categories were similarly sensitive to half-lives, whereas titer and minimum categories were less sensitive to resistance fractions. The ratio does not strongly predict any outcomes relative to variation in all other parameters, a sign that there is no general solution to optimizing the ratio and it must be adjusted on a case-by-case basis.

interaction were distinct. The remaining outcomes were correlated but further categorized as neutralization, $\log_{10}$ titer, coverage metrics (% of viruses neutralized > 99%), and IIP. Results were similar for AUC and trough, see **S2 Fig**. As the model used a monotonic PK curve, the final value was roughly representative of the entire time-course.

## Correlations among PKPD outcomes and antibody features

We next explored the associations between a representative member of each outcome category and model parameters (**Fig 2B**). All categories were sensitive to PK (half-life), and generally more to the half-life of the shorter-lived bNAb (hl2). Increased resistance negatively correlated with the outcomes, particularly with neutralization, $\log_{10}$-transformed titer, coverage metrics, and IIP. Additionally, a stronger negative correlation was found with the resistant fraction for the bNAb with longer half-life–this pattern was weaker for mean IC50. Total dose correlated positively with all outcomes but was generally less influential than other model parameters. The ratio of antibodies did not strongly predict any outcome after accounting for variation in all other parameters, highlighting that there was no generally optimal ratio; optimization is determined on a case-by-case basis based on many antibody features.

## Sensitivity of the optimal ratio for each outcome

Next, for each parameter set, we determined the optimal ratio *r* for each outcome. **Fig 3A** shows an analogous clustering analysis to **Fig 2** but with correlations of the *optimal ratio* of each outcome across the inputs. Importantly, the same general categories emerged such that correlations among all outcomes agreed generally with correlations among optimal ratios. However, for the BH and additive interactions, the $\log_{10}$ titer and IIP now cluster together. Additionally, the minimum interaction model clustered into 2 categories based on the potency and coverage outcomes. Interestingly, the optimal ratio for minimum (non-coverage) is often negatively correlated to the others. This suggests that optimizing for minimum interaction (i.e., maintaining consistent combination coverage) may require a very different ratio. For the other interactions, once an outcome is selected, the optimal ratios generally agree among the additive and Bliss-Hill interaction models.

   **Fig 3B** shows correlations among model parameters and optimal ratios from a representative outcome from each category. In this plot directionality of correlation has additional meaning: positive and negative correlations imply less or more of the antibody with worse half-life, respectively. The sensitivity to PK and viral PD inputs (complete resistance and mean IC50) followed the same pattern as in **Fig 2A**: all the outcomes showed some sensitivity to PK, titer and minimum interaction outcomes were sensitive to mean IC50, and the remaining were sensitive to resistance fractions. For ratio optimization, the PK sensitivity was specifically driven by the half-life of the shorter-lived bNAb.

   We next sought to understand what is gained by using the optimal ratio as opposed to a more practical solution near the optimum. Therefore, we measured how many parameter combinations admitted an outcome within 95% of the outcome value achieved by the optimal ratio (**Fig 3C**). That is, if many simulations were within 95% of the optimum, it means the optimum is not substantially better. Indeed, for the parameter ranges we considered, some outcomes were not particularly sensitive to the choice of the optimal ratio such that other practical considerations could be promoted in a trial design. However, some outcomes were much more strongly affected by optimization (with fewer than $1/10^4$ runs being in the 95% optimal scenario). So, although there are cases of insensitive systems (e.g., two poor products, two highly effective products), this reinforces that optimization should be case-specific.

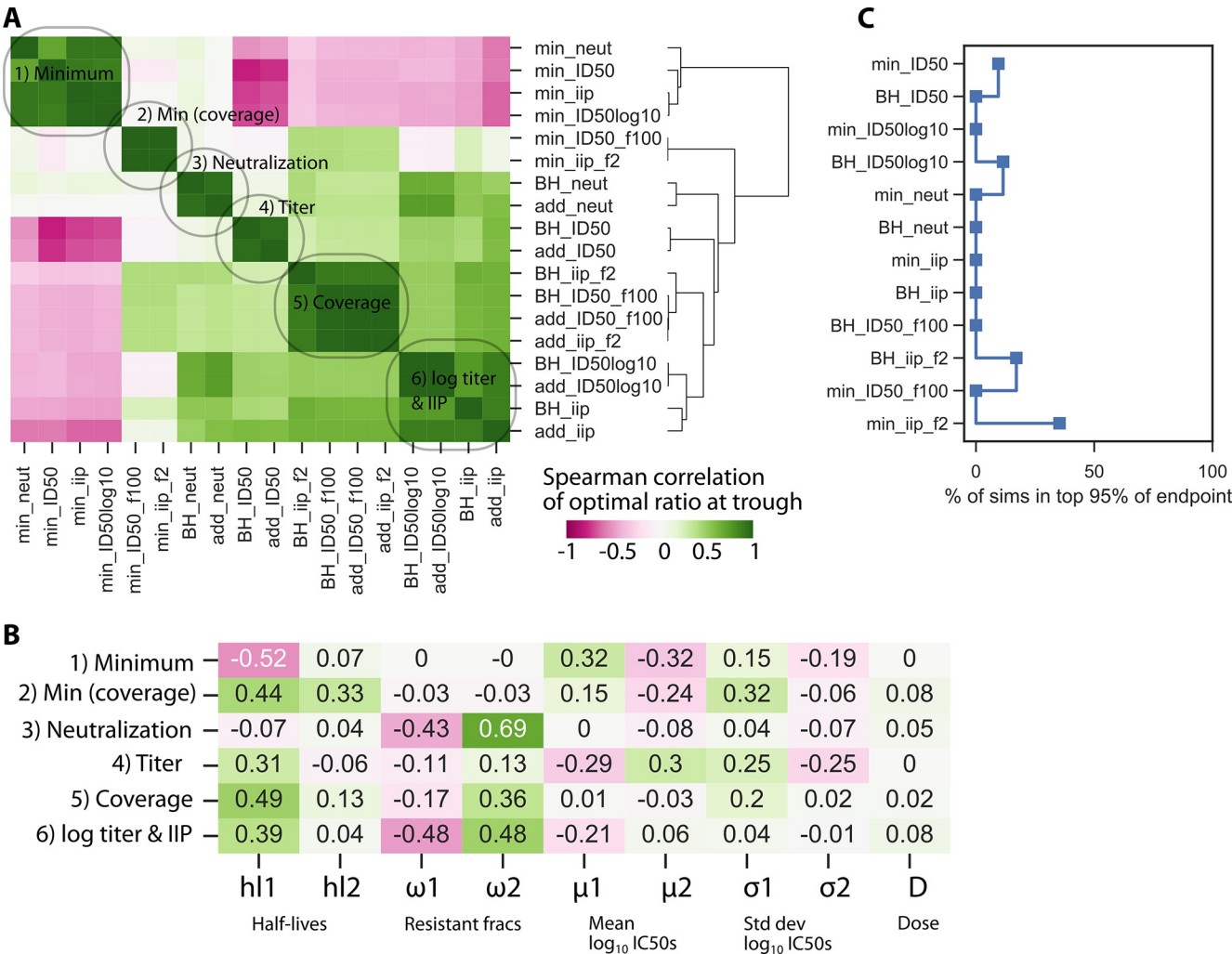

**Fig 3. Sensitivity of the optimal ratio to PKPD outcome choices and antibody features.** The optimal ratio was calculated for each parameter set and each outcome. A) Optimal ratios cluster by outcome similarly to the general analysis in **Fig 2** with differences being that coverage and IIP can be grouped together whilst the minimum interaction model separates into 2 categories. B) Different variables drive optimization of different outcomes. Among viral pharmacodynamic inputs, the geometric mean IC50 is most influential on titer and minimum potency outcomes, while the fraction resistant is most influential for the remaining outcomes. C) A low percentage of parameter sets admitted outcomes within 95% of the optimal value.

## Dual parental antibodies outperformed the bispecific product without synergy enhancement

Bi-specific antibodies, synthesized combinations of two antibodies into one product, exhibit superior neutralization compared to their parental components *in vitro* [20,21]. However, biochemical properties (e.g., molecular weight, valence, structure) change when parental Abs are synthesized into a single unit) [22]. Whether these changes influence *in vivo* pharmacokinetics remains to be seen. For example, the clinical candidate 10e8/iMab [NCT03875209] has excellent HIV neutralization [21,23], but includes ibalizumab (iMab), which has complex PK with fast clearance (effective half-life < 1 week) [FDA Biologic License Application 761065]. In that context, we sought to test a worst-case scenario in which a bi-specific inherits the worse (faster) parental half-life. We assumed two parental antibodies: one with a 3-month half-life and mediocre potency and another with higher potency but short (1 week) half-life. We compared the

bi-specific version inheriting the best potency but worst half-life with theoretical combinations of the drugs separately using a realistic design administering 300 or 1200 mg of total bNAbs over a 3-month administration window and the following PKPD outcomes: a continuous outcome (mean IIP) and a coverage outcome (% viruses IIP>2).

Compared to the combination therapy, the superior potency of the bi-specific antibody was not always sufficient to counterbalance its short half-life. Across doses and interaction models, we consistently found that the optimal combination therapy was more efficacious than the bi-specific for both AUC and trough (**Fig 4**). At trough, where half-life is strongly influential, the longer half-life parent with worse PD dosed by itself still outperformed the bi-specific.

Synergy has been observed for bi-specifics because binding of one antibody arm can facilitate the second to bind [24]. Given this finding, we tested how much additional synergy (as a factor multiplying the bi-specific potency through reduced IC50, see **Methods**) could rescue the bi-specific performance and make it comparable to the parental combination. The bi-specific outperformed the optimized combined administration when the synergy factor exceeded 10-fold under common interaction models (**Fig 4**).

## Incorporating empirical protection correlates in clinical design

To perform a realistic optimization of a clinical trial, we consider deviations from *in vitro* potency that may be relevant for *in vivo* protection. For example, non-human primate SHIV challenge studies suggest that a bNAb titer of approximately 100 achieved 50% protection: i.e., serum antibody concentrations need to be 100-fold higher than *in vitro* IC50 to elicit 50% protection *in vivo* [14]. We define the fold-increase as a "potency reduction factor" [13], $\rho$, and henceforth translate *in vitro* potency to *in vivo* protection by scaling the titer input. We have *in vivo* neutralization and IIP then,

$$v^{in\ vivo}(t) = \left\{1 + [\tau_{ij}^{in\ vitro}(t)/\rho]^{-1}\right\}^{-1},$$

Eq 4

$$IIP_{ij}^{in\ vivo}(t) = \log_{10}[1 + \tau_{ij}^{in\ vitro}(t)/\rho].$$

Eq 5

such that no change from *in vitro* measured titer occurs when $\rho = 1$ and a potency reduction of 100-fold means $\rho = 100$. Mechanistically, this formulation suggests that the overestimated protection *in vivo* is due to either (or both) underestimation of the potency due to some biological factors (e.g., coagulation or anti-antibody elements) or overestimation of bNAb concentration at the site of exposure.

The reduction factor can be derived from assessing actual protection at the given experimental titers, either through NHP challenge data [14] or using protection efficacy (PE) estimated from the antibody mediated prevention (AMP) studies testing the bNAb VRC01 for prevention of HIV in humans [5]. The titer vs. protection dose-response relationship may also be derived using alternative functional forms than those depicted Eqs 4 and 5. We derived dose-response relationships from both a meta-analysis of NHP challenge studies and the AMP studies (see **Methods**). Fitting both the potency reduction model and a 5-parameter logistic (5PL) model, we find the more flexible 5PL model better captures the slope and asymmetry exhibited in the empirical protection estimates (**S3 Fig**). Results from the AMP studies also suggest a larger potency reduction ($\rho = 370$) than what was predicted by the NHP meta-analysis ($\rho = 91$).

For combination bNAbs, the experimental titer will represent neutralization in sera with a combined concentration of antibody. Whether a potency reduction factor is applied to the combination titer or to the individual titers prior to the interaction is specifically consequential

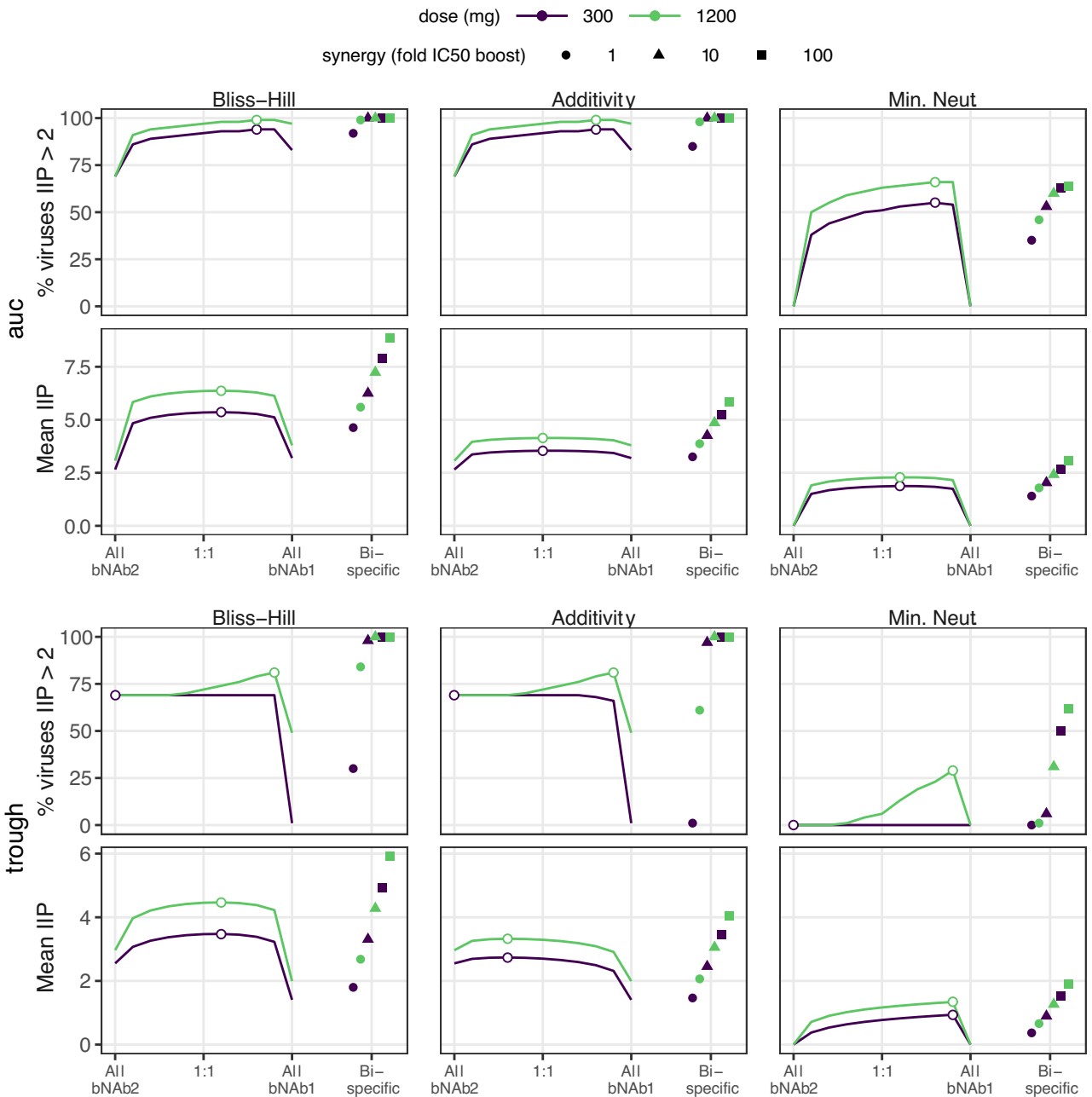

**Fig 4. Optimizing 2 bNAb combination therapy in comparison to bi-specific therapy with the same bNAbs.** Combination antibody results for AUC (top) and trough (bottom) suggest that trough is slightly more sensitive to ratio (see curvature of outcome surface and change from optimal ratio denoted by open dot). In general, a single bi-specific bNAb will perform worse than combination therapy if it has the best neutralization potential of both parental lineages under a common interaction model but inherits the faster clearance kinetics. However, if synergetic binding occurs, enhancing the bi-specific potency by 10-fold (see **Methods**), it is similar or outperforms the optimal combination for all outcomes and doses. "All bNAb1" and "All bNAb2" on the x-axis correspond to 100% dosing of the second bNAb product.

for the Bliss-Hill interaction model, but not the other interaction models. Briefly, applying the factor to the Bliss-Hill combination titer model may be overly conservative, underestimating the protection because experimental titer does not uniquely predict Bliss-Hill neutralization (**S4 Fig**; see **S1 Text** for further discussion). We suggest applying the potency factor or

protection model to each bNAb individually, calculating their individual protection estimate, then applying the Bliss-Hill interaction model (i.e., at the event-level) as described in the following case study.

### Using empirical protection correlates in a 3-bNAb optimization

We gathered several independent data sets to model a 12-week trial with a 600 mg subcutaneous dose of 3 state-of-the-art broadly neutralizing antibodies (3BNC117-T, 10-1074-T, VRC07-523-LS; -T denotes theoretical variant with extended half-life). For this example, we compared empirical protection estimates based on titer using a meta-analysis(14) and the AMP studies(5) (**S3 Fig**). We optimized for PKPD target outcomes, using both AUC and trough, of viral coverage at 50% and 95% protection thresholds. For more details on the input PK and PD for these analytes, see **Methods** and **S5 Fig.** In this illustrative example, we do not consider clade-specific profiles nor account for interference potentially due to 3BNC117 and VRC07-523-LS targeting the same epitope (CD4-bs). We tested all double and triple combinations varying the dosing ratios. In a practical clinical setting, complicated dosing ratios (e.g., 98:13:3) might not be ideal. Thus, for the 3-bNAb combination, we considered simple ratio designs: an even dose split (denoted 1:1:1) or any 50%:25%:25% combination (denoted 2:1:1 or similar). We compared this to the theoretical optimum to ensure they were reasonably close to the optimal design. The optimal ratio always contained <10% 3BNC117-T but varied otherwise depending on the target PKPD outcomes and empirical protection correlate usually favoring >50% VRC07-523-LS (**S2 Table**).

Overall, protection levels were uniformly lower when predicted using the AMP clinical study results vs. the NHP meta-analysis. Using the NHP results, all triple drug combinations predicted a protection level above 95% for roughly 25% of viruses at trough or on average (AUC) (**Fig 5**). Likewise, protection levels were above 50% for roughly 75–80% of viruses over the study. Using the AMP results, generally <50% of viruses were protected at levels above 50% in this design, and coverage generally tracked only slightly higher than the 95% threshold level of the NHP results. It was clear that VRC07-523LS was the best single antibody, and the optimal dosing ratio generally contained >60% of VRC07-523-LS (see **S2 Table**) with one exception: to protect at 95% levels using the AMP protection estimate, the best design requires a majority of 10-1074-T. Optimal ratios were generally consistent across target outcomes whether using AMP or NHP empirical protection estimates, but there were exceptions, specifically when protection was worsening, highlighting the complex interplay between PK and PD under the design constraints. Subsequently, the triple combination with 1:1:2 level of VRC07-523LS was not much worse than optimal, even the outcome where 10–1074 is optimal. Moreover, the optimal 2-drug combination without 3BNC117-T was generally as effective as the optimal 3 drug therapy (which dosed at <10% of 3BNC117-T) potentially due to general lower potency of 3BNC117 or the overlap in epitope targeting with VRC07-523-LS resulting in redundant viral coverage in the database. Still, given our necessarily incomplete data on circulating strains, we would suggest using this 3-drug regimen at a 1:1:2 design to balance simplicity and protection for this example.

### Discussion

Combination administration of broadly neutralizing antibodies will likely be tested in coming studies of HIV prevention and cure [1,4,21,25]. Therefore, we developed an approach to define optimal combinations in terms of the ratio of bNAbs in a combination administration (e.g., 1:1:1 would be an equal ratio in a triple bNAb combination).

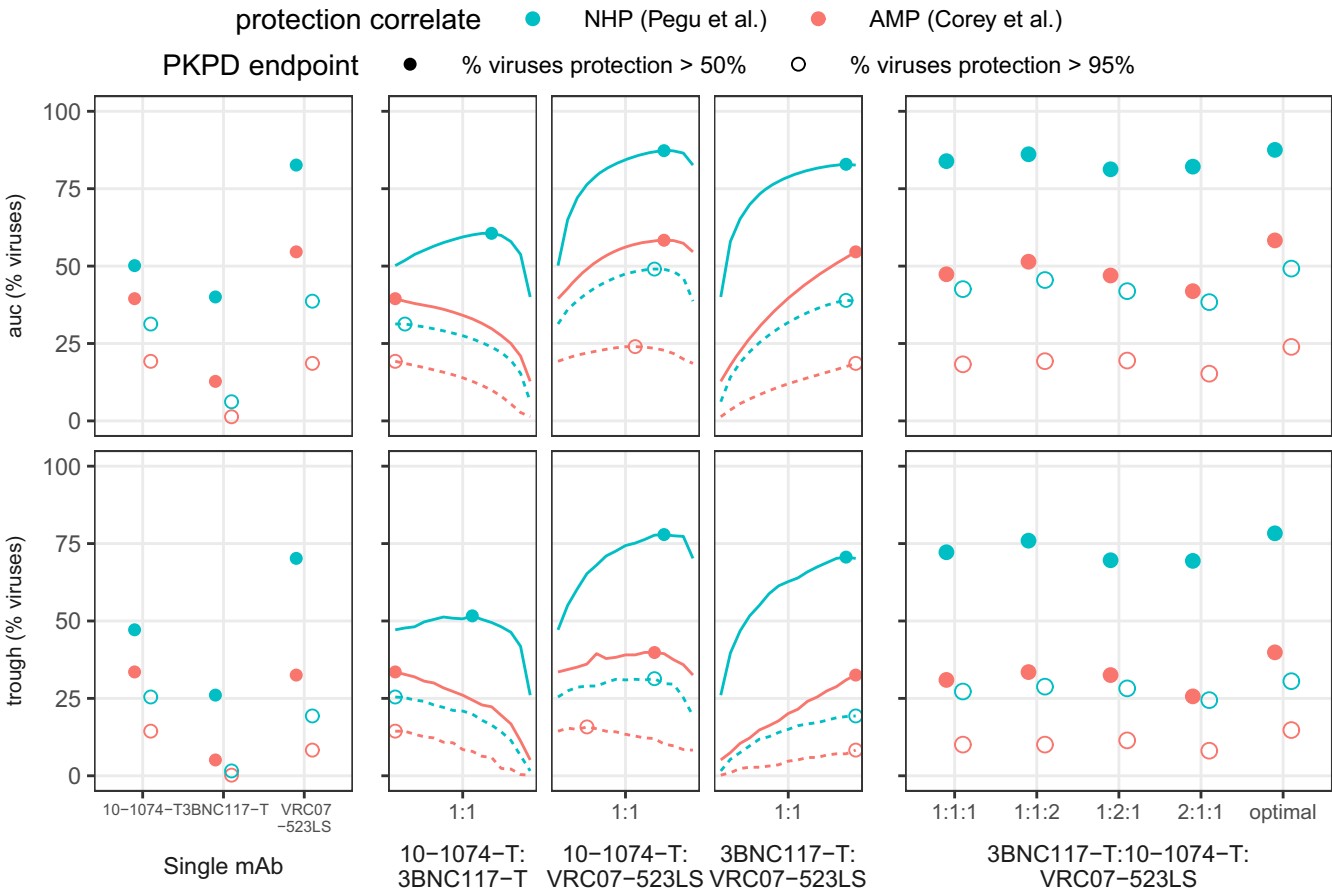

**Fig 5. Additional enhancement after optimization of 3-drug therapy.** Using 3 well known anti-HIV broadly neutralizing antibodies, we performed an analysis predicting the percent of viruses covered at more than 50% and 95% protection levels using protection correlates from an NHP meta-analysis(14) and the AMP clinical trials(5). The percent viruses covered were computed over the total time course using area under the curve (auc) and at the final time point (trough). We compared coverage for the bNAbs individually, in dual combination, and in triplicate as 1:1:1, 1:1:2, 1:2:1, 2:1:1, and the optimal combination (see **S2 Table**). Enhancement over the best single bNAb (VRC07-523-LS) is generated through combinations when evaluating the percent of the viruses neutralized at a 95% level. However, triple drug therapy does not meaningfully enhance over optimized 2-drug therapy levels, even when completely optimized. Protection levels are more optimistically predicted using the NHP meta-analysis vs. the AMP trials (see also **S3 Fig**), and optimal designs can depend on the underlying protection correlate (e.g., 10-1074-T:VRC07-523LS auc). Indeed, a 1:1:1 3 drug therapy is outperformed by the optimized 2-drug therapy, highlighting the need to carefully perform case-studies for any optimization scenario.

Optimal antibody ratios balance neutralization against bNAb longevity and also depend on how combinations interact *in vivo*. Therefore, several types of data for each antibody in a combination modality must all be considered to optimize dosing. These include 1) *in vitro* assay data relating each bNAb's potency (IC50) against many HIV variants; 2) *in vivo* bNAb pharmacokinetics, 3) a correlate that can translate *in vitro* IC50 to *in vivo* protection; and 4) any relevant data on bNAb interactions. Our present approach addresses how to optimize combination bNAb dosing by integrating all 4 data types.

Available *in vitro* data includes pseudovirus panels (e.g., CATNAP database [17]) or breakthrough viruses in human infections [26]. Alternative approaches may be desirable to augment these data. For example, data on HIV-1 Env sequences are abundant, so modeling precision could be improved using new techniques to predict IC50s from Env sequences [27–29].

Correlate data is particularly crucial because *in vitro* IC50 measurements typically overestimate *in vivo* efficacy [13,30]. By comparing correlates derived from NHP and human studies, we identified that optimal dosing was sensitive to the correlate value. A key feature of our

approach was the derivation (**S3 Fig**) and implementation of such a correlate. As in our prior work [13], we used a "potency reduction factor" to mathematically model the discrepancy between *in vitro* and *in vivo* IC50s.

Our work emphasizes that protection estimates derived from single bNAb studies need to be carefully translated into combination optimization. We showed that for the favored Bliss-Hill (BH) interaction, combination titer does not uniquely specify protection: it is possible to arrive at the same combination titer with different underlying bNAb concentrations. We instead propose defining antibody-level potency, applying the interaction, and then mapping to clinical protection. This procedure would also be amenable to additional complexities that we did not address here including potency reduction factors and/or correlates that differ by bNABs in a combination.

We analyzed a theoretical bi-specific motivated by the clinical candidate 10e8/iMab [NCT03875209], which has excellent neutralization [21,23], but includes ibalizumab (iMab) that has complex PK with fast clearance (effective half-life < 1 week) [FDA Biologic License Application 761065]. This example (**Fig 4**) illustrated how clearance kinetics (PK) could offset strong neutralization (PD). However, adding beneficial synergy (plausible based on co-binding) made the bi-specific outperform the combination therapy. In general, bi-specifics are encouraging—without synergy or a difference in PK, bi-specifics provide roughly a 2-fold concentration bonus compared to a 1:1 administration of the parentals at an equivalent dose. They may be clinically preferable as they are a single product. Yet, co-formulation of multiple products carries manufacturing considerations (e.g., changes in viscosity, introductions of interference) that are hard to predict with respect to PK and neutralization. A final practical issue that emerged was that enough synergy in the model made the bi-specific potent even at concentrations below current limits of detection. Though we modeled bi-specifics, recent work with tri-specific antibodies shows they can prevent SHIV infection [31], but may depend on their strongest component [32].

One potent antibody can also determine the ability of combinations. Indeed, in the triple combination example in **Fig 5**, a two-drug regimen would have been nearly as good. Albeit minorly, adding the third antibody and optimizing the triple-drug ratio was always better than the best two-drug combination. Thus, in considering that viral panels are necessarily incomplete, we would err on the side of inclusivity to both widen breadth and as a hedge for unknown escape processes.

The precise endpoint to optimize for HIV prevention or therapy remains a question of opinion. Therefore, we showed there are categories that are co-optimized by the same dose ratios (**Fig 3**). Whereas additive and maximum interactions are highly similar, the correct definition of additive or BH interaction may still improve optimization. On the other hand, the minimum interaction formed a unique cluster of simulated endpoints and admits different optimal ratios. While the minimum interaction is not biologically favored, in practice it might still be considered to cover a worst-case scenario where there is no protection against viruses without sensitivity to at least two bNABs. Additionally, certain endpoints were more sensitive to optimal dosing than others, a finding that could be considered in endpoint selection. Or alternatively, if an endpoint is preferred which is not particularly sensitive to ratio, practical considerations about dosing could be prioritized over precise dose optimization.

Although most of our analysis concerns prevention studies, this framework is applicable to curative interventions attempting to use bNABs to prevent viral rebound after stopping ART [33,34]. However, blocking a single founder during a transmission event appears easier than blocking repeated reactivations of diverse viral populations from latent reservoirs. Although, levels required to prevent rebound remain hard to predict, several studies have demonstrated bNABs can delay viral rebound [33–35]. The *in vivo* potency reduction factor appears large in

this context, in a cohort of 18 individuals receiving VRC01 infusion and ART cessation, rebound occurred while plasma VRC01 levels were still well above *in vitro* IC50s [34].

Going forward, our recommendation for designing therapeutic combinations for prevention or treatment of diverse pathogens is several fold: 1) choose outcomes based on expert opinions and given disagreements, assess whether these qualitative decisions are actually quantitatively in agreement; 2) consider multiple, distinct outcomes to evaluate a range of potential results; 3) optimize drug ratios for the specifics of component features; and 4) include subdominant levels of weaker antibodies to potentially cover holes in coverage not observable from incomplete preliminary data.

## Methods

### Code and data

All analysis were performed in R and Python. Simulations, data processing, and visualizations performed using R used the *tidyverse* package suite [36]. Sensitivity and cluster analysis of simulation results with subsequent visualizations were performed using the seaborn library in Python. All code will be available on GitHub.

### Estimation of Hill slope using CATNAP data

The Hill slope in the 2-parameter logistic Hill function (**Eq 2**) can be estimated from the IC50 and IC80 measurements (formula derived the **S1 Text**). We estimated the distribution of the neutralization Hill slope by performing this calculation across virus/antibody combinations available in the LANL CATNAP database(17). To accommodate assay quantification limits that potentially vary across experimental study, we limited the analysis datasets to IC50 and IC80 values between 0.01 and 20 ug/mL, comprising 20,236 total combinations. Additionally, we grouped calculations within quartiles of input IC50 to assess whether Hill slopes vary by underlying viral sensitivity or measurement error that varies with the scale of IC50.

### Global sensitivity analysis

We performed ~10,000 simulations over all combinations of parameters in Table 2 and calculated all PKPD outcomes. We chose a one-compartment exponential PK model with trough time 84 days for each bNAb: $C_i(t) = C_i(0) \exp(-k_i t)$, and summarized the PK model with its half-life $hl_i = \ln 2/k_i$. The PK model used a one compartment model with initial condition defined as the dose scaled by a volume of distribution ($C_i(0) = D/V$) fixed to 3 L based on previous studies of human IgGs [37,38]. One bNAb was simulated to always have equivalent or better half-life than the other to avoid redundancy. We chose a log-normal distribution for IC50s for each bNAb parameterized by its mean $\mu_i$ and standard deviation $\sigma_i$ on the $\log_{10}$ scale, also allowing for a fraction $\omega_i$ that are completely resistant (infinite IC50). We sampled 500 viruses per simulation. We then varied these parameters, along with the ratio of doses $r$ and the total dose $D$. Then, we determined the optimal ratio as the ratio that maximized each PKPD outcome for all other parameter values across interaction models. To calculate IIP under Bliss-Hill, we used neutralization for each bNAb as the basis input as described in Table 1 (see **S1 Text** for explanation on why IIP is not calculated using combined BH titer). Using the seaborn package in Python, we performed hierarchical clustering of Spearman correlations among outcomes and between parameters and outcomes.

## Comparison of bi-specific to parental antibodies

For the first parental bNAb, we chose a potent neutralizer (mean IC50 of $10^{-3}$ with 0% viral resistance) but with poor PK: elimination half-life equivalent to 1/12 of the administration period (i.e., 7-day half-life for an 84-day trough). For the second bNAb, we chose a more modest neutralizing profile (mean IC50 of $10^{-2}$ with higher variance and 33% viral resistance) but with excellent PK: elimination half-life equivalent to one administration period.

To model the bi-specific, we assumed the single molecule formulation means two parental products are given at the identical dose. We also assumed the clearance PK was determined by the faster of the two parental products. We additionally allowed for synergy, such that each antibody's potency is improved by a factor $\alpha$. This factor was assumed to be the same for all viral strains. Thus, following **Eq 3** and **Table 2**, the bi-specific IIP against a single virus $V_j$ can be calculated for max, min, and additivity models, respectively

$$\mathrm{IIP_j} = \log_{10}[1 + \alpha \ \min_i \tau_{ij}], \tag{Eq 6}$$

$$\mathrm{IIP_j} = \log_{10}[1 + \alpha \ \max_i \tau_{ij}], \tag{Eq 7}$$

$$\mathrm{IIP_j} = \log_{10}[1 + \alpha \sum_i \tau_{ij}]. \tag{Eq 8}$$

For Bliss-Hill interaction, the derivation from individual titers to a combination IIP is shown in the **S1 Text** and then the bi-specific synergy was implemented as follows:

$$\mathrm{IIP_j} = \sum_i \log_{10}[1 + \alpha \tau_{ij}]. \tag{Eq 9}$$

For comparing the combination and bi-specific therapies, we examined IIP and % viruses having IIP>2 (a surrogate of protection in nonhuman primate studies [14]) for AUC and trough. Calculations were based on 500 simulations as implemented for the global sensitivity analysis.

## Deriving titer vs. protection dose-response relationships

We tested several models to map *in vivo* protection from *in vitro* neuralization using both NHP meta-analysis data and the AMP clinical trial results[5,14]. In the NHP single high dose challenge study, the authors developed a logistic regression model to predict protection probability from *in vitro* neutralization titer [14]. We use the titer of their model at 50%, 75%, and 95% protection as input for our model. A dose-response relationship was established between the IC80 against VRC01 and the prevention efficacy (PE, using data depicted in Fig 3 of Ref. [5]). To derive a titer relationship, we assumed the IC80 dose-response relationship depicts the per-exposure infection probability for the average treated AMP participant when exposed to a given virus (i.e., IC80). Under the simplifying assumption that infection time is random, we estimated the average concentration of a treated participant by taking the weighted mean of the midpoint concentrations across the VRC01 study groups (36.2 mcg/mL, Table 1 Ref. [5]). The weights were the group sample size over the overall sample sizes. We then translated the IC80 to ID80 titer simply by dividing IC80 by the average concentration, and then translated ID80 to the ID50 assuming a Hill slope of 1 (see **S1 Text**).

To derive the dose-response relationship, we employed the following approach: for a given bNAb ($i$) at a given concentration, we estimated *in vivo* protection ($p$) using neutralization titer ($\tau_{ij}$) against a virus ($j$). Using **Eq 3** and estimating a single parameter, the potency reduction, was $\rho = 1/91$ for the NHP data and 1/370 for the AMP data. For each, a better fit was

achieved using a 5-parameter logistic (5PL) model, a generalized dose-response type function with 5 parameters {*A,B,C,D,E*} and the form

$$y(x) = D + (A - D)\{1 + \exp[B(\log(x) - \log(C))]\}^{-E},$$ Eq 10

here mapping *in vitro* titer $x = \tau_{ij}$ to *in vivo* protection $y = p_{ij}$. We fixed $D = 0\%$ and $A = 100\%$ so that protection ranges from 0–100%. For the NHP data, the remaining 3 parameters were estimated as $B = -1.84$, $C = 257$, and $E = 0.338$; and for the AMP data, the remaining 3 parameters were estimated as $B = -1.86$, $C = 404$, and $E = 0.892$. The best fit of the potency reduction model and the 5PL model are compared in **S3 Fig**.

## Realistic clinical trial simulation

The full trial design contained a 12-week observation window and 600 mg total subcutaneous (SC) dosing with PK parameters established from clinical study (**S1 Table** and **S5A Fig**). To boost performance of 3BNC117 and 10–1074, we artificially enhanced their half-lives by 3-fold to mimic an -LS variant (3BNC117-T and 10-1074-T). The distribution of *in vitro* neutralization against circulating strains was modeled using *in vitro* derived IC50s from 507 available common strains in the LANL CATNAP database(17) (**S5B Fig**).

We then illustrate predictions of the 5PL models using the NHP and AMP protection estimates (**S3 Fig**) for each bNAb comparing *in vitro* neutralization to *in vivo* protection via % viral coverage >50 and >95% in **S5C Fig**. Using this model of protection, we then calculated combined protection across the administered bNAbs (*b* is the number of antibodies considered) assuming independence similar to Bliss-Hill:

$$p_j = 1 - \prod_i^b (1 - p_{ij})$$ Eq 11

We then defined our protection PKPD outcome as viral coverage fraction such that we can determine what % of all viruses have protection above a certain threshold value *X*:

$$f(t, X) = \frac{1}{n} \sum_{j=1}^n \mathcal{I}(p_j(t) > X)$$ Eq 12

where $\mathcal{I}$ is the indicator function equal to 1 if the inequality holds and 0 otherwise.

We assessed PKPD at the trough time (12-weeks, T) and as an average over the administration period (AUC/T over time through T).

## Supporting information

**S1 Text. Additional methodological text and derivations.**
(PDF)

**S1 Fig. Estimated neutralization Hill slope using CATNAP data.** Estimated Hill slopes for different ranges of IC50 measurements (split into quartiles). Each point represents a calculation for an antibody/virus combination in the database where IC50 and IC80 measurements were both within 0.01–20 ug/mL range. Median Hill slope estimates (IQR) displayed above each box plot. There were approximately 5000 measurements per quartile.
(EPS)

**S2 Fig. Additional clustering results for AUC.** As for trough in **Fig 2A**, endpoints cluster by Spearman correlation into similar 5 main categories, from top to bottom: titer, minimum, additive titer, neutralization/coverage, and IIP.
(TIF)

**S3 Fig. Empirically estimated dose-response between titer and protection.** Relationship between neutralization titer (concentration divided by IC50) and *in vivo* protection using two different data sets: 1) NHP challenge meta-analysis titers (blue) that predict 50%, 75%, and 90% protection, and 2) AMP clinical trials data (red) extracted from the AMP IC80 vs. protection efficacy curves and converted into titer (see **Methods** in main text). Expected *in vitro* neutralization at given titer shown as a dashed black line, potency reduction depicted via curve shift achieving 50% protection shown as dotted line (multiplicative shift factor, *p*, listed in legend), and 5PL model curve fitted over empirical protection estimates (points on curves) depicted via solid lines.
(EPS)

**S4 Fig. IIP and ID50 relationship by interaction model against a single virus at varying concentration and bNAb ratios.** For the virus, bNAb1 had an IC50 of 1 and bNAb2 had an IC50 of 0.1. Predicted experimental titer is a combination titer (i.e., dilution factor applied to sera containing both antibodies). A) The predicted experimental ID50 titer by interaction model (see **Table 1** for $ID_{50}$ titer formulas) across total bNAb concentrations at two ratios of the individual bNAb (1:1 and 1:10 denoted by colors). B) The predicted neutralization IIP by interaction model (see **Table 1** and for neutralization formulas) across total bNAb concentrations and ratios. IIP was calculated as the $\log_{10}$-transformation of one minus the combined neutralization. For Bliss-Hill, the lines cross at increasing concentration indicating the 1:1 ratio performs better, a qualitatively different conclusion than **A**. C) Relationship between predicted experimental ID50 titer and neutralization across total bNAb concentrations and ratios. The black line indicates the predicted relationship between IIP and ID50 for single antibody/virus combinations. For additivity, all lines overlap indicating the relationship holds. For Bliss-Hill, separate lines indicate that the experimental ID50 does not correspond to a unique neutralization. For example, at a predicted BH experimental ID50 of 100, the 1:1 ratio elicits the highest neutralization, and both ratios elicit higher neutralization than predicted by using titer.
(EPS)

**S5 Fig. Input data for 3-bNAb combination optimization.** A) PK over 12 weeks for each of the three antibodies given at a 600mg dose using subcutaneous route. B) Neutralization data for each of the products from 507 viruses in CATNAP database. Distribution of IC50s among sensitive viruses (IC50 < 10) are shown as box plots in the bottom plot and bar plots depict the total resistant viruses (IC50 > 10) in the top plot. C) For increasing concentrations for the individual antibodies, the percent of viruses sensitive at given thresholds (50% and 95%) and concentrations (x-axis). The first column depicts neutralization coverage when concentration exceeds the IC50 values depicted in **B**. For increasing concentrations, the second two columns depict protection coverage based on estimates from an NHP meta-analysis and the AMP clinical trials as calculated in **S3 Fig**.
(EPS)

**S1 Table. Population PK input parameters for the empirical case study optimization.**
(DOCX)

**S2 Table. Ratio optimization results from 3-bNAb optimization for 3BNC117-T, 10-1074-T, and VRC07 523-LS.**
(DOCX)

## Author Contributions

**Conceptualization:** Bryan T. Mayer, Daniel B. Reeves.

**Data curation:** Bryan T. Mayer, Allan C. deCamp, Yunda Huang.

**Formal analysis:** Bryan T. Mayer, Daniel B. Reeves.

**Funding acquisition:** Raphael Gottardo, Peter B. Gilbert, Daniel B. Reeves.

**Investigation:** Bryan T. Mayer, Allan C. deCamp.

**Methodology:** Bryan T. Mayer.

**Project administration:** Daniel B. Reeves.

**Resources:** Allan C. deCamp, Yunda Huang.

**Software:** Bryan T. Mayer.

**Supervision:** Yunda Huang, Joshua T. Schiffer, Raphael Gottardo, Peter B. Gilbert, Daniel B. Reeves.

**Validation:** Daniel B. Reeves.

**Visualization:** Bryan T. Mayer, Daniel B. Reeves.

**Writing – original draft:** Daniel B. Reeves.

**Writing – review & editing:** Bryan T. Mayer, Joshua T. Schiffer, Peter B. Gilbert.

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
