## [Decision Letter · Decision Letter 0]

1 Dec 2021

Dear Dr Reeves,

Thank you very much for submitting your manuscript "Optimizing clinical dosing of combination broadly neutralizing antibodies for HIV prevention" for consideration at PLOS Computational Biology.

As with all papers reviewed by the journal, your manuscript was reviewed by members of the editorial board and by several independent reviewers. In light of the reviews (below this email), you are invited to submit a revised paper; however, there was a range of comments, some quite critical, which need to be adequately addressed in a resubmission.

We cannot make any decision about publication until we have seen the revised manuscript and your response to the reviewers' comments. Your revised manuscript is also likely to be sent to reviewers for further evaluation.

Sincerely,

James Gallo

Associate Editor

PLOS Computational Biology

Thomas Leitner

Deputy Editor

PLOS Computational Biology

Reviewer's Responses to Questions

**Comments to the Authors:**

Reviewer #1: Mayer et al provide an important framework for considering what ratio of bNAbs within a multi-bNAb cocktail is optimal, given a single dose of fixed size. The model builds upon prior work in the field and incorporates PK, PD and interactions between bNAbs under multiple interaction scenarios. Useful case-studies comparing dose considerations for a bispecific vs 2-bNAb combiantion, and consideration of various bNAb ratios within a clinically relevant 3-bNAb cocktail, are illustrative. The analyses appear very well developed and comprehensive in scope. The author’s insights, in addition to their publicly available web-based tool for ratio optimization, should benefit investigators when planning future clinical or preclinical studies to evaluate prophylactic or therapeutic combinations of bNAbs.

Addressing the following minor comments should help improve the clarity and readability of the manuscript:

Line 44: The sentence beginning “Less sensitive variants . . .” is unclear and appears to state the opposite of what was stated two sentences earlier.

Line 46: typo (PrEP)

Line52: Please check accuracy of reference 12. Also, the Schiffer, et al reference is listed twice in the bibliography (as ref 12 and 26)

Lines 449-450: Please rephrase the sentence beginning “To calculation IIP under Bliss-Hill . . .” for clarity.

Reviewer #2: 1. This framework adds exceptionally to the field and enables design of future combination antibody studies. Furthermore, this tool can be used to enable more robust phase 2/3 study designs.

2. PK modelling for the AMP trial was pivotal for dose selection which predicted the target bNAb concentrations. Unfortunately, the assumptions about sensitivity/early resistance in the population were off target. It would be great to see examples of how the authors use the clinical data as an input into the system to enable the readers to see the comparison of how the pre-clinical data versus the actual clinical data translates

3. The authors highlight practical considerations about dosing. It is important to address the other complexities and challenges with the use of combination products. A single co-formulated drug product is ideal but will also need to take into account the specific formulations etc. It would be nice to have additional characteristics such as viscosity and concentration formulations inputted into the system to determine if the products can be co-formulated as well.

4. The authors mention limitations “although most of our analysis concerns prevention studies, this framework is applicable to curative studies attempting to use bnAbs to prevent viral rebound after stopping ART-within-host diversity in the reservoir is the challenge here”

Previously described literature suggests use of predictive modelling based on env sequencing of the individual’s quasispecies to determine bnAb sensitivity patterns and optimize combination cocktails, including scoring AA signature types for bnAb resistance, prior to using a particular bnAb. Refer to Magaret CA, Benkeser DC, Williamson BD, Borate BR, Carpp LN, Georgiev IS, et al. Prediction of VRC01 Neutralization Sensitivity by HIV-1 Gp160 Sequence Features. PloS Comput Biol (2019) 15:e1006952. doi: 10.1371/journal.pcbi.1006952

Have the authors considered adding these features to the existing model?

Reviewer #3: The manuscript uses theoretical model to evaluate the effect of combination of monoclonal antibodies on the outcome of anti-HIV therapy. The work being evaluated is very relevant and mathematical modeling and simulation can provide great insight into the goal. However, unlike authors’ previous work (i.e., ref # 18 and 19) the work presented here is purely speculative, and add little to the field. Either most conclusions are obvious, or the analysis is inclusive. In addition, the author use 1-compartment PK model with 3L volume to capture antibody PK, which is bound to provide higher than actual contractions. Also, the assumption that bi-specific antibody would have shorter half-life is not well-founded. Rather than simulative 4 different interaction conditions, may be in vitro work can be used to choose most realistic nature of combination. Lastly, there needs to be some comparison of model simulation with realistic data (e.g., clinical PK of mAb, viral dynamic data etc.) to provide confidence in the model and its output. Lastly, the manuscript is written in a way that the reader can get lost very easily if not familiar with authors’ previous work. It may be worth rewriting some part of the paper.

**Have the authors made all data and (if applicable) computational code underlying the findings in their manuscript fully available?**

Reviewer #1: Yes

Reviewer #2: Yes

Reviewer #3: Yes

PLOS authors have the option to publish the peer review history of their article (what does this mean?). If published, this will include your full peer review and any attached files.

Reviewer #1: No

Reviewer #2: No

Reviewer #3: No
---

## [Decision Letter · Decision Letter 1]

8 Mar 2022

Dear Dr Reeves,

We are pleased to inform you that your manuscript 'Optimizing clinical dosing of combination broadly neutralizing antibodies for HIV prevention' has been provisionally accepted for publication in PLOS Computational Biology.

Best regards,

James Gallo

Associate Editor

PLOS Computational Biology

Thomas Leitner

Deputy Editor

PLOS Computational Biology

Reviewer's Responses to Questions

**Comments to the Authors:**

Reviewer #1: The authors have addressed all of my comments.

Reviewer #2: In my opinion, the authors have addressed all responses adequately. This framework adds exceptionally to the field and enables design of future combination antibody studies. Furthermore, this tool can be used to enable more robust phase 2/3 study designs. Congratulations to the team.

**Have the authors made all data and (if applicable) computational code underlying the findings in their manuscript fully available?**

Reviewer #1: Yes

Reviewer #2: Yes

PLOS authors have the option to publish the peer review history of their article (what does this mean?). If published, this will include your full peer review and any attached files.

Reviewer #1: No

Reviewer #2: **Yes: **Sharana Mahomed

---

## [Editor Report · Acceptance letter]

30 Mar 2022

PCOMPBIOL-D-21-01795R1 

Optimizing clinical dosing of combination broadly neutralizing antibodies for HIV prevention

Dear Dr Reeves,

I am pleased to inform you that your manuscript has been formally accepted for publication in PLOS Computational Biology. Your manuscript is now with our production department and you will be notified of the publication date in due course.

With kind regards,

Agnes Pap
